# Oral Administration of Deer Bone Collagen Peptide Can Enhance the Skin Hydration Ability and Antioxidant Ability of Aging Mice Induced by D-Gal, and Regulate the Synthesis and Degradation of Collagen

**DOI:** 10.3390/nu16111548

**Published:** 2024-05-21

**Authors:** Ke Zhang, Chenxu Zhao, Kaiyue Liu, Ruyi Feng, Yan Zhao, Ying Zong, Rui Du

**Affiliations:** 1College of Traditional Chinese Medicine, Jilin Agricultural University, Changchun 130118, China; wojiaozhangkea@163.com (K.Z.); zs6607582@gmail.com (C.Z.); 15870540735@163.com (K.L.); fdd16639230916@163.com (R.F.); zhyjlu79@163.com (Y.Z.); 2Jilin Provincial Engineering Research Center for Efficient Breeding and Product Development of Sika Deer, Jilin Agricultural University, Changchun 130118, China; 3Key Laboratory of Animal Production and Product Quality and Safety, Ministry of Education, Jilin Agricultural University, Changchun 130118, China

**Keywords:** deer bone, collagen peptide, hydration, antioxidation, synthesis and degradation of collagen

## Abstract

Skin problems caused by aging have attracted much attention, and marine collagen peptides have been proved to improve these problems, while mammalian collagen peptides are rarely reported. In this study, fermented deer bone collagen peptide (FCP) and non-fermented deer bone collagen peptide (NCP) were extracted from fermented and non-fermented deer bone, respectively, and their peptide sequences and differential proteins were analyzed using LC-MS/MS technology. After they were applied to aging mice induced with D-gal, the skin hydration ability, antioxidant ability, collagen synthesis, and degradation ability of the mice were studied. The results show that FCP and NCP are mainly peptides that constitute type Ⅰ collagen, and their peptide segments are different. In vivo experiments show that FCP and NCP can improve the richness of collagen fibers in the skin of aging mice; improve the hydration ability of skin; promote the activity of antioxidant-related enzymes; and also show that through the TGF-β and MAPK pathways, the synthesis and degradation of collagen in skin are regulated. These results show that deer bone collagen peptide can improve skin problems caused by aging, promote skin hydration and antioxidant capacity of aging mice, and regulate collagen synthesis and degradation through the MAPK pathway.

## 1. Introduction

Aging is an irreversible mechanism of cell cycle arrest, and it is a process that leads to the general decline in the function of every organ and tissue of the body through complex and multifaceted influences, but its specific causes and mechanisms are still unknown [1,2]. At present, there are nine recognized signs of aging: gene instability, telomere decrease, epigenetic changes, loss of protein homeostasis, nutritional sensory imbalance, mitochondrial dysfunction, cell aging, stem cell failure, and changes in intercellular communication [3,4,5]. Since it was first described in vitro in 1961, aging has always been a problem that people want to explore. There are even different opinions on the causes of aging, and there is still no unified theory of aging. However, the reasons supported by many scholars include: telomere shortening, DNA damage, inflammatory response, and immune response of the body [6,7,8].

At present, many scholars have studied the problem of aging, and the methods of establishing aging model are: D-gal induction [9], hydrogen peroxide induction [10], ultraviolet induction [11], and so on. Among them, the aging model induced with D-galactose is a widely used method, which simulates many biological changes related to human aging [12]. The D-galactose-induced aging model shows similar characteristics to human aging at molecular and cellular levels, such as increased oxidative stress [13,14], shortened telomere length [15,16], DNA damage [17,18], and increased apoptosis [19,20]. In addition, the D-galactose-induced aging model is based on the principle of metabolic disorders, which simulates the metabolic abnormalities in the aging process by continuously administering D-galactose to the animals, and it is comparable to the aging process in humans [21,22,23]. The D-galactose-induced aging model based on metabolic disorders and increased oxidative stress, and these factors also play an important role in the skin aging process. Therefore, this model is very suitable for studying collagen degradation, elastic fiber degeneration, skin thickness reduction, and changes in the antioxidant defense system caused by skin aging. This is exactly in line with the needs of this paper, so this paper chooses D-gal to establish the skin aging model, and according to the modeling method of Jing et al. [24], D-gal was injected continuously intraperitoneally at a dose of 200 mg kg^−1^ for 8 weeks to induce skin aging in mice.

Collagen peptide is a hydrolysate of collagen, which has the advantages of smaller molecular weight and easier absorption compared to collagen [25]. Collagen is a biomacromolecule protein synthesized by animal cells. Collagen is the most abundant protein in mammals, accounting for 25~35% of the total protein [26]. At present, 28 kinds of collagen have been found, which are widely distributed in animal tissues such as skin, bones, blood vessels, muscles, and so on [27]. Collagen can be hydrolyzed to obtain a variety of bioactive peptides, which have the characteristics of low molecular weight, easy absorption, high bioavailability, prototype resistance, and hypoallergenicity, and have been widely used in functional foods, skin care products, biomedical products, and other fields.

With the increase in age and the stimulation of environmental factors, collagen and water in the skin are lost, and oxidative stress and glycosylation modification continue to occur, resulting in dry and rough skin, loss of elasticity, and increased wrinkles, which are all manifestations of skin aging [28]. Collagen peptide can reduce the accumulation of ROS in cells, up-regulate the activities of the antioxidant-related enzymes SOD, CAT, and GSH-Px, and reduce the content of MDA, a product of lipid peroxidation, thus improving antioxidant damage [29,30]. Liang et al. [31], found that long-term oral administration of salmon collagen peptide can promote the synthesis of collagen Ⅰ and Ⅲ, down-regulate the expression of matrix metalloproteinase-1(MMP-1) to inhibit the degradation of collagen, and finally maintain the steady state of collagen in skin. In addition, the collagen peptide of Pacific cod skin can inhibit the activation of activator protein 1(AP-1) by down-regulating the mRNA levels of c-Jun and c-Fos, thus inhibiting the expression and activity of MMPs, finally reducing the degradation of collagen and effectively preventing skin photoaging caused by ultraviolet radiation [32].

In this study, we extracted collagen peptides from fermented and non-fermented deer bones and analyzed their peptide sequences and differential proteins using the LC-MS/MS technique. Then, we carried out the study of deer bone collagen peptides using ELISA, RT-qPCR, Western Blot, and other experiments. The effects of deer bone collagen peptides on aging-induced skin changes were illustrated in terms of the hydration, antioxidation, and regulation of collagen synthesis and degradation. Thus, it provides a theoretical basis for the development and utilization of deer bone.

## 2. Materials and Methods

### 2.1. Materials, Chemicals, and Animals

Sika deer bones are purchased from the market and obtained by removing meat, tendons, and other impurities. *Lactobacillus acidophilus* was purchased from BNCC Biotechnology Co., Ltd. (Beijing, China). Healthy KM mice (20 ± 2 g, five weeks old) were purchased from Yisi Experimental Animal Technology Co., Ltd. (Changchun, China) with permit number: SCXK(JI)2023-0002. Human skin fibroblasts (HSF) were purchased from 51cells Co., Ltd. (Wuhan, China). Superoxide dismutase (SOD), catalase (CAT), glutathione peroxidase (GSH-Px), and malondialdehyde (MDA) kits were purchased from Nanjing Jiancheng Bioengineering Corporation, Ltd. (Nanjing, China). Hydroxyproline (HYP), hyaluronic acid (HA), total antioxidant capacity (T-AOC), mouse collagen type Ⅰ (COLⅠ), mouse collagen type Ⅲ (COLⅢ), mouse transforming growth factor β(TGF-β), and mouse matrix metalloproteinase 1(MMP-1) enzyme-linked immunoassay (ELISA) kits were purchased from Youxuan Biotechnology Co., Ltd. (Shanghai, China). All animal experiments have been approved by the welfare and ethics committee of experimental animals of Jilin Agricultural University (acceptance number of ethics review: 20211011003).

### 2.2. Preparation of Collagen Peptide (CP) from Deer Bone

Collagen peptide from deer bone was extracted according to the preliminary research in the laboratory. Deer bone was mixed with sterile water at the ratio of 1:26 (g/mL), inoculated with *Lactobacillus acidophilus* at the inoculation amount of 12% (*v*/*v*), and fermented under anaerobic conditions at 37 °C for 34.5 h. The fermented deer bone was extracted with hot water (95 °C, 3 h) at the ratio of material to liquid of 1:10 (g/mL), and the filtrate was freeze-dried. Then, the freeze-dried powder was dissolved until the substrate concentration was 4%, and was hydrolyzed using trypsin (6300 U/g, 37 °C, 4 h), and the supernatant was freeze-dried. At the same time, the above operations were carried out on unfermented deer bones. Finally, two kinds of deer bone collagen peptides (CP) were obtained, which were fermented deer bone collagen peptide (FCP) and non-fermented deer bone collagen peptide (NCP).

### 2.3. LC-MS/MS Analysis

Separation of the samples was carried out with the HPLC liquid system. Buffer A was an aqueous solution of 0.1% formic acid, and Solution B was 0.08% formic acid (acetonitrile 80%). Balance the column with 100% of the solution A and load the sample into a precolumn of mass spectrometry (C18 3 μm 100 μm × 20 mm, Thermo Scientific, Waltham, MA, USA) before separating it with an analytical column (C18 1.9 μm 150 μm × 120 mm, Thermo Scientific).

The samples were separated using capillary HPLC and analyzed using an Orbitrap Fusion Lumos mass spectrometer (Thermo Scientific, Waltham, MA, USA). Parameter setting: analyzing time 78 min, detecting method of positive ion, scanning range of parent ion is 300~1400 *m*/*z*, primary mass spectrogram is 120,000, AGC target is 5 × 10^5^ primary maximum IT is 50 ms, and Doppler exclusion time is 20.0 s. Polypeptide and its mass-to-charge ratio are obtained according to the following method: MS2 Activation Type is HCD, Isolation window is 1.6 *m*/*z*, Microscans is 1, Secondary Maximum IT is 35 ms, and Normalized Collision Energy is 33 eV.

### 2.4. Quantitative Analysis of Differential Proteins, GO Function Annotation, and KEGG Pathway Annotation

Under the threshold condition of 2 difference multiples (FC = fold change), the differential proteins were screened. FC ≥ 2 means up-regulation, FC ≤ 0.5 means down-regulation, and 0.5 < FC < 2 means that the expression level has no significant change.

Use EBI database and InterProScan v5.31-70.0 software to annotate all differentially expressed proteins with GO function, and at the same time, count the number of differential proteins at the level of GO secondary function annotation.

The annotation of proteins was analyzed through the KEGG pathway database.

### 2.5. Cell Viability Determination

Human skin fibroblasts (HSF) in the logarithmic growth phase were inoculated into 96-well plates at a density of 1 × 10^5^ cells/mL, and cultured at 37 °C and 5% CO_2_ for 12 h. FCP and NCP were diluted to solutions with concentrations of 15.625, 31.25, 62.5, 125, 250, 500, and 1000 μg/mL, respectively, and were added into 96-well plates, at the same time, a blank control group was set up, and each sample was provided with 3 multiple wells, and the culture was continued for 24 h. Add 10 μL CCK-8, and incubate for 40 min at 37 °C with 5% CO_2_ in the dark. After incubation, the absorbance at 450 nm was measured using the enzyme-labeled instrument, and the cell proliferation rate was calculated.

### 2.6. Animal Experiment Grouping and Drug Administration

Seventy SPF Kunming mice aged 5 weeks, half male and half female, weighing 20 ± 2 g, were subjected to temperature (24 ± 1 °C) and humidity (60 ± 5%), and 12 h bright environment and 12 h dark environment were created alternately. All mice were able to eat and drink freely. After one week of acclimatization, they were randomly divided into 7 groups (10 mice per group):(I)Normal control group (NC): normal saline.(II)Model group (D-gal): D-galactose; normal saline.(III)Positive drug group (VC): D-galactose; VC (dose: 400 mg·kg^−1^ body weight).(IV)Fermented deer bone collagen peptide high dose group (FH): D-galactose; FCP (dose: 400 mg·kg^−1^ body weight).(V)Fermented deer bone collagen peptide low dose group (FL): D-galactose; FCP (dose: 200 mg·kg^−1^ body weight).(VI)Non-fermented deer bone collagen peptide high dose group (NH): D-galactose; NCP (dose: 400 mg·kg^−1^ body weight).(VII)Non-fermented deer bone collagen peptide low dose group (NL): D-galactose; NCP (dose: 200 mg·kg^−1^ body weight).

D-galactose treatment was performed by intraperitoneal injection of 0.2 mL of D-galactose (dose: 200 mg kg^−1^ body weight) every day for 8 weeks (56 days). The mice were given 0.2 mL of CP or normal saline by gavage at one hour after treatment with D-galactose.

After the last dose, the mice were anesthetized by intraperitoneal injection of 10% chloral hydrate at a dose of 0.1 mL 10 g^−1^. Blood was taken from the eyeball of mice, and the whole blood was allowed to stand at room temperature for 2 h, centrifuged at 3000 rpm for 10 min, and the supernatant was taken to obtain a serum sample for subsequent experiments. After anesthesia, the hair on the mouse’s back was shaved with a hair scraper, and the remaining fine villi were removed with depilatory cream to expose smooth skin. After eyeball blood was taken, the skin of the area about 2 cm × 2 cm on the back of the mouse was cut off with scissors, and fascia and fat were carefully peeled off to obtain mouse skin tissue for subsequent experiments.

### 2.7. Histological Staining of Skin

The mouse skin tissue were fixed for 24 h in 4% paraformaldehyde solution, dehydrated, paraffin embedded, and cut. Skin sections were stained with hematoxylin and eosin (H&E) and stained with hematoxylin, ponceau, and aniline blue (MASSON). Observe under a microscope.

### 2.8. Oxidative Stress, Hyp, and HA Content in Skin and Serum

Skin tissue were homogenized in an ice bath using a tissue homogenizer with nine-fold normal saline (*w*/*w*), and then centrifuged for 15 min at 12,000× *g* and 4 °C, taking the supernatant to obtain skin tissue homogenate. The skin tissue homogenate and serum were treated according to the methods described in the instructions of each kit, and the levels of superoxide dismutase (SOD, A001-1-2, Nanjing Jiancheng, China), catalase (CAT, A007-1-1, Nanjing Jiancheng, China), glutathione peroxidase (GSH-Px, A005-1-2, Nanjing Jiancheng, China), total antioxidant capacity (T-AOC, YX-200118M, Shanghai Youxuan, China), and malondialdehyde (MDA, A003-1-2, Nanjing Jiancheng, China) in the skin tissue homogenate and serum were detected. At the same time, the content of hyaluronic acid (HA, YX-080100M, Shanghai Youxuan, China) and hydroxyproline (HYP, YX-080907M, Shanghai Youxuan, China) in the skin tissue homogenate was detected.

### 2.9. ELISA

The serum obtained in 2.6 and the skin tissue homogenate obtained in 2.8 were used for ELISA detection. According to the steps described in the instructions of each kit, finally, the absorbance was detected using an enzyme-labeled instrument (Synergy HTX, BioTek) to obtain the content of collagen type Ⅰ (COLⅠ, YX-031521M, Shanghai Youxuan, China), collagen type Ⅲ (COLⅢ, YX-031539M, Shanghai Youxuan, China), transforming growth factor β (TGF-β, YX-200708M, Shanghai Youxuan, China), and matrix metalloproteinase 1 (MMP-1, YX-131317M, Shanghai Youxuan, China) in mouse skin and serum.

### 2.10. RT-qPCR

Total RNA was extracted from mouse skin using the TRNzol Universal kit (Tiangen Biotech Co., Ltd., Beijing, China). Reverse transcription of the total RNA into cDNA was carried out in accordance with the instructions using the PrimeScript™ FAST RT reagent kit (Takara Bio Co., Ltd., Beijing, China) according to the instructions. Using the TB Green^®^ Fast qPCR Mix (Takara Bio Co., Ltd., Beijing, China) to carry out the qPCR according to the instructions.

### 2.11. Western Blot

Western blot was used to analyze the expression of the MAPK pathway. The skin tissues of seven groups of mice, including the normal control group (NC), model group (D-gal), and administration group (VC, FH, FL, NH, NL), were, respectively, cut and then added to lysis solution (mixed with protease inhibitor 100:1 before use), fully ground with a tissue homogenizer, stood on ice for 15 min, and centrifuged at 4 °C 12,000 rpm for 15 min. Take the supernatant to obtain the total protein of skin tissue. The BCA kit was used to quantify the total protein in each group of skin tissues. Then, a loading buffer was added and boiled in boiling water for 15 min, and finally protein samples were obtained. The samples were separated using SDS-PAGE, then transferred to the PVDF membrane, sealed for 2 h with 5% skimmed milk, incubated overnight at 4 °C with suitable primary antibodies (ERK:1:10,000, Wuhan Sanying; JNK:1:5000, Wuhan Sanying; p38:1:3000, Wuhan Sanying; c-jun:1:1000, Wuhan Sanying; c-fos:1:5000, Wuhan Sanying; GAPDH:1:100,000, Wuhan Sanying), washed with TBST, incubated for 2 h with the secondary antibody, and washed with TBST finally developing.

### 2.12. Statistical Analysis

The experimental data are expressed with mean standard deviation (Mean ± SD), and the data are statistically analyzed using SPSS Statistics 26. The mean values between samples are compared using one-way analysis of variance (ANOVA) followed by Tukey’s post hoc test for group comparisons, and the image is analyzed using Image J 2.14.0 software, with *p* < 0.05 being statistically significant.

## 3. Results

### 3.1. LC-MS/MS

#### 3.1.1. Identification of Collagen Peptides from Deer Bone by Mass Spectrometry

As shown in Figure 1, 45 proteins and 231 peptides were identified from FCP, and 22 proteins and 153 peptides were identified from NCP; see Appendix A for details. The distribution of peptide length contained in both is shown in Figure 2, which is mainly concentrated between 7 and 29 peptide bonds.

Figure 3 shows two peptides with the strongest signal in FCP and two peptides with the strongest signal in NCP, respectively. The two peptide sequences in FCP are: GETGPAGPAGPIGPVGAR and SGDRGETGPAGPAGPIGPVGAR, both of which are identified as the peptide segments that constitute the α1 chain of type I collagen. The two peptide sequences in NCP are: RGETGPAGPAGPIGPVGAR and GEVGPAGPNGFAGPAGAAGQAGAK. After identification, the former is the peptide segment that constitutes the α1 chain of type I collagen, and the latter is the peptide segment that constitutes the α2 chain of type I collagen.

#### 3.1.2. GO Function Annotation

Quantitative analysis of differential proteins between the two samples showed that, compared with NCP, FCP had 173 up-regulated proteins and 76 down-regulated proteins. The GO function annotation of differential proteins is shown in Figure 4, which involves 13 biological processes: biological regulation, cellular process, localization, metabolic process, cellular anatomical entity, protein-containing complex, ATP-dependent activity, binding, catalytic activity, cytoskeletal motor activity, molecular function regulator activity, structural molecule activity, and transporter activity. Among them, the three biological processes with the largest number of differential proteins are: binding, cellular anatomical entity, and structural molecule activity. 

#### 3.1.3. KEGG Path Annotation

The KEGG pathway annotation of differential proteins was carried out, and the results are shown in Figure 5. The main signal pathways involved in differential proteins include: motor proteins, protein digestion and absorption, focal adhesion, human papillomavirus infection, ECM-receptor interaction, PIK-Akt signaling pathway, adrenergic signaling in cardiomyocytes, cardiac muscle contraction, dilated cardiomyopathy, hypertrophic cardiomyopathy, and so on.

### 3.2. Effect of CP on the Viability of HSF Cells

As shown in Figure 6A, compared to the blank control group, when the concentration of FCP is in the range of 15.625~1000 μg/mL, the cell viability increases with the increase in FCP concentration. When the concentration was 500 μg/mL, the cell viability reached the maximum (*p* < 0.01), and if the concentration continued to increase, the cell viability began to decline. It can be seen that FCP has no cytotoxic effect on HSF when the concentration is lower than 1000 μg/mL, and the activity of HSF cells is dose-dependent with the concentration of FCP in the range of 15.625~500 μg/mL.

Similarly, as shown in Figure 6B, compared with the blank control group, when the concentration of NCP is in the range of 15.625~1000 μg/mL, with the increase in NCP concentration, the cell viability increases. When the concentration was 500 μg/mL, the cell viability reached the maximum (*p* < 0.01), and if the concentration continued to increase, the cell viability began to decline. It can be seen that NCP has no cytotoxic effect on HSF when the concentration is lower than 1000 μg/mL, and the activity of HSF cells is dose-dependent with the concentration of FCP in the range of 15.625~500 μg/mL.

### 3.3. Histological Staining of Skin

#### 3.3.1. H&E Staining

As shown in Figure 7, the skin structure of the normal control group is complete, the hair follicles are abundant and evenly distributed, and the epidermis and dermis are closely connected. Compared to the normal control group, the model group has fewer visible hair follicles, reduced dermal thickness, and a loose and disorderly arrangement of collagen fiber bundles, which has typical aging pathological damage. Compared to the model group, the hair follicles in the skin of the drug group are more complete in structure, more in number, evenly arranged, thicker in dermis, and richer in collagen fiber bundles, which indicates that the drug group can effectively alleviate the skin changes of aging mice induced with D-gal.

#### 3.3.2. MASSON Staining

As shown in Figure 8, the blue-dyed collagen fibers in the skin of the normal control group are concentrated and dense, and the collagen fibers are closely connected, with high collagen content and elastic skin. In the model group, the components of blue-stained collagen fibers distributed in the dermis of skin tissue decreased, and the blue-stained collagen fibers were sparsely distributed and loosely connected with each other. Compared to the normal control group, the collagen fibers in the model skin lost their compactness and showed atrophy. Compared to the model group, the compactness of blue-dyed collagen fibers in the skin of the treatment group increased, and the collagen fibers were closely connected, not dispersed, and the collagen was closely distributed, which effectively improved the aging state of collagen fibers in the skin.

### 3.4. Effect of CP on Skin Hydration in Mice

HYP is the main amino acid in collagen, which has strong hydrophilicity. The decrease in its content will lead to the decrease in water content and water retention in skin. HA is an acidic mucopolysaccharide, which widely exists in tissues and organs such as the dermal layer and articular cartilage. It has high hygroscopicity and good moisture retention, and it is an ideal natural moisturizing factor. Therefore, the content of HYP and HA in the skin and serum of aging mice were detected using ELISA to evaluate the hydration ability of mouse skin.

As shown in Figure 9A, the HYP and HA levels in the skin and serum of the model group were lower than those in the normal control group (*p* < 0.05). Compared with the model group, both FCP and NCP can increase the levels of HYP and HA in the skin and serum of aging mice (*p* < 0.05).

Filaggrin, involucrin, and loricrin are three natural skin moisturizing factors, and their expressions are closely related to the formation of the epidermal skin barrier and the maintenance of skin moisture. Therefore, the expression of filaggrin, involucrin, and loricrin genes were detected using RT-qPCR.

Results as shown in Figure 9B. Compared to the normal control group, intraperitoneal injection of D-gal significantly reduced the expression of filaggrin, involucrin, and loricrin genes in the skin of model mice (*p* < 0.05). Compared to the model, FCP and NCP can significantly up-regulate the expression of filaggrin, involucrin, and loricrin genes in the skin of aging mice (*p* < 0.05).

The results showed that FCP and NCP could up-regulate the content of HYP and HA in the skin and serum of aging mice, and promote the expression of filaggrin, involucrin, and loricrin genes, thus improving the decline in hydration ability of mouse skin caused by aging, and the effect of FCP on skin hydration of aging mice was better than that of NCP (*p* < 0.05).

### 3.5. Effect of CP on Skin Antioxidant Capacity of Aging Mice

The ELISA method was used to determine the effects of CP on the antioxidant indexes of the skin and serum of aging mice. As shown in Figure 10, the MDA content in the skin and serum of the model group was significantly higher than that in the normal control group (*p* < 0.05); SOD, CAT, and GSH-Px were significantly reduced (*p* < 0.05), and the total antioxidative capacity was significantly reduced (*p* < 0.05).

Compared to the model group, FCP can significantly reduce the content of MDA in the skin and serum of aging mice (*p* < 0.05), promote the activities of SOD, CAT, and GSH-Px in the skin and serum of aging mice (*p* < 0.05), and significantly enhance the total antioxidant capacity (*p* < 0.05). Among them, the activities of CAT and GSH-Px in the serum of aging mice were not significantly improved.

Compared with the model group, the high-dose NCP can obviously reduce the content of MDA in the skin and serum of aging mice (*p* < 0.05), and enhance the activities of SOD, CAT, and GSH-Px and the total antioxidant capacity (*p* < 0.05). The low-dose group can significantly reduce the content of MDA in the skin and serum of aging mice (*p* < 0.05), and increased the activities of SOD in the serum and CAT and GSH-Px in the skin (*p* < 0.05) and total antioxidant capacity in the skin and serum (*p* < 0.05).

The results showed that CP could reduce the accumulation of MDA in serum and tissues of aging mice, promote the activity of antioxidant-related enzymes, and enhance the total antioxidant capacity, thus playing an antioxidant role, and FCP had a better antioxidant effect than NCP (*p* < 0.05).

### 3.6. Effect of CP on Collagen Production Ability of Aging Mice Skin

The content of COLⅠ, COLⅢ, MMP-1, and TGF-β in the skin and serum of aging mice and the expression of mRNA were detected. The results are shown in Figure 11. Compared to the normal control group, the content of COLⅠ, COLⅢ, and TGF-β in the skin and serum of the model group decreased (*p* < 0.05), while the content of MMP-1 increased (*p* < 0.05).

Compared to the model group, FCP high and low dose groups can significantly increase the content of COLⅠ, COLⅢ, and TGF-β in the skin and serum of aging mice (*p* < 0.05), reducing the content of MMP-1 (*p* < 0.05); RT-qPCR results showed that FCP could significantly increase the expression of COLⅠ, COLⅢ, and TGF-β mRNA in the skin of aging mice (*p* < 0.05) and decrease the expression of MMP-1 mRNA (*p* < 0.05), which was consistent with the results of ELISA.

Compared to the model group, the NCP high-dose group can significantly increase the content of COLⅠ, COLⅢ, and TGF-β in the skin and serum of aging mice (*p* < 0.05), reducing the content of MMP-1 (*p* < 0.05). The low-dose NCP group can significantly increase the levels of TGF-β in the skin and COLⅠ and TGF-β in the serum of aging mice (*p* < 0.05), and decrease the level of MMP-1 in the serum of aging mice (*p* < 0.05). RT-qPCR results showed that NCP could significantly increase the expression of COLⅠ, COLⅢ, and TGF-β mRNA in the skin of aging mice (*p* < 0.05), and decreased the expression of MMP-1 mRNA (*p* < 0.05), and the results were consistent with those of ELISA.

The results showed that CP could promote the expression of the TGF-β protein and gene, and up-regulate the synthesis of collagen Ⅰ and Ⅲ. At the same time, it inhibits the expression of the MMP-1 protein and gene, thus down-regulating the degradation of collagen Ⅰ and Ⅲ, and then promoting the synthesis of collagen in aging mouse skin. The ability of FCP to promote collagen synthesis was significantly better than that of NCP (*p* < 0.05).

### 3.7. Effect of CP on the Expression of MAPK Signaling Pathway Related Proteins

As shown in Figure 12, compared with the normal control group, the expressions of three subtypes of the MAPK pathway ERK, JNK, and p38 in the model group were significantly enhanced (*p* < 0.05), the expressions of c-jun and c-fos, which constitute the AP-1 complex, were significantly up-regulated (*p* < 0.05). Compared to the model group, each treatment group can reduce the expression of ERK, JNK, and p38 in the skin of aging mice, and inhibit the expression of c-jun and c-fos protein in a dose-dependent manner. It shows that CP can regulate the expression of ERK, JNK, and p38, inhibit the activation of the MAPK signaling pathway, and then down-regulate the expression of its downstream factors c-jun and c-fos, reduce the formation of the AP-1 complex, down-regulate the content of MMP-1, and finally inhibit the degradation of collagen, and the effect of FCP is better than that of NCP.

## 4. Discussion

Aging is an inevitable process of the human body with the growth of age, and its most intuitive result is manifested in the skin [33]. As one of the first lines of defense for human immunity, skin is always exposed to various stimuli in the environment [34]. Stimulations such as ultraviolet (UV) radiation, particle adhesion, and temperature change in the external environment, combined with oxidative stress and glycosylation in the body, all change the appearance, structure, and function of the skin, resulting in decreased skin elasticity, increased wrinkles, relaxation, and abnormal pigmentation [35]. The intake of D-galactose can produce advanced glycation end products (AGEs), which lead to oxidative stress. Oxidative DNA damage will activate the MAPK pathway, activate transcription factor-activated protein-1(AP-1), promote the production of MMP-1, and lead to the degradation of collagen. Therefore, we used D-gal as a modeling drug to establish a skin aging model using the intraperitoneal injection of a high concentration D-gal into Kunming mice [9]. After 8 weeks, we observed that the hair in the model group was sparse and that hair loss was aggravated, which was consistent with the description in the literature, indicating that the model was successful.

Hydroxyproline (HYP) and hyaluronic acid (HA) are important natural moisturizing factors in human skin, which can combine with water molecules, and the combined water molecules have good stability, thus improving the water content and elasticity of skin [36,37]. Filaggrin, involucrin, and loricrin are important proteins related to the epidermal barrier, which can form keratin hyaline granules in the granular layer and keratinized films in keratinocytes, thus maintaining skin barrier function and reducing transepidermal water loss (TEWL) [11,38,39,40]. Our study found that the content of HYP and HA in the skin of aging mice increased, the expressions of filaggrin, involucrin, and loricrin mRNA were up-regulated, and the hydration ability of the skin of mice was improved. The results were consistent with those of Kim et al. [41]. Improving the hydration ability of the skin can keep the skin moist, reduce the generation of wrinkles, and improve the roughness of the skin, thus playing an anti-aging role.

SOD, CAT, and GSH-Px are important antioxidant enzymes in the body, which can reduce hydrogen peroxide and eliminate superoxide anion free radicals [42]. MDA is the product of lipid peroxidation, and the increase in its content indicates that oxidative stress occurs in the body [43]. As markers of oxidative stress, detecting the levels of SOD, CAT, GSH-Px, and MDA can measure the degree of oxidative stress [44]. We found that oral CP can improve the activity of antioxidant-related enzymes and inhibit the production of MDA. This is consistent with the results of Chen et al. [45] and Liang et al. [46], and it shows that CP can inhibit the oxidative stress process of skin. Oxidative stress is one of the most important causes of skin aging, so CP may be able to delay skin aging through antioxidation.

With an increase in age, the collagen fibers in the skin are lost and the elasticity is weakened, so wrinkles will appear, the skin will become rough, and the skin will become aged [47]. Therefore, promoting the synthesis of collagen is very important to delay skin aging. TGF-β is an important participant in the process of tissue repair, which can promote the production of a line of collagen [48,49], while MMP-1 can degrade collagen [50,51]. Our study found that CP can up-regulate the level of TGF-β and down-regulate the level of MMP-1 in the skin, thus regulating the generation and degradation of collagen and increasing the content of collagen in the skin. At the same time, the same result can be obtained by staining the skin tissue of mice: the collagen fiber bundles in the skin of aging mice are more abundant, and are arranged neatly after taking CP orally.

The extracellular stimuli that the MAPK pathway can transmit include biological stress stimuli such as cytokines and neurotransmitters, and abiotic stresses such as osmotic pressure, oxidative stress, and DNA damage. There are four subtypes of the MAPK pathway: ERK, JNK, p38, and ERK5. The ERK MAPK pathway is the classic pathway of MAPK, which can form the Ras-Raf-MEK-ERK pathway and regulate the downstream c-Fos factor [52]. The JNK MAPK pathway and p38 MAPK pathway are similar in function, so they are usually considered to be the JNK/p38 MAPK pathway together [53,54]. After activation, they enter the nucleus and combine with the target gene of c-jun, thus affecting the gene expression [55]. AP-1 is a heterodimer, which consists of different families of proteins, including c-Fos, c-Jun, and ATF [56]. At present, research shows that c-Fos and c-Jun are genes significantly related to aging. After being activated in the MAPK pathway, they combine to form the AP-1 complex, and combine with their receptors in MMPs initiation region to regulate the expression of MMP-1, improve the degradation of type I collagen fibers, and make the skin relax and wrinkles deepen [57,58,59]. Therefore, inhibiting the signal transduction of the MAPK signaling pathway and down-regulating the expression of MMP-1 are the key mechanisms that play an anti-aging role. Our experimental results show that oral CP can inhibit the activation of the MAPK pathway and down-regulate the expression levels of c-fos and c-jun proteins, thus inhibiting the degradation of collagen. This is consistent with Liu et al. [60] and Kim et al. [61].

Using LC-MS/MS analysis, we identified 266 peptides from FCP and NCP, and the length of peptide segments mainly concentrated between 7~29 peptide bonds. Among them, the peptide segment with the largest proportion is the peptide segment on the α1 chain of type I collagen, followed by the peptide segment on the α2 chain of type I collagen; that is to say, FCP and NCP are mainly the products of the hydrolysis of type I collagen. The low metabolic stability and bioavailability of most peptides are important reasons that limit the development of peptide oral products [62,63,64]. The proteolytic enzymes in the gastrointestinal tract and serum, as well as metabolic enzymes in the liver, pose a serious threat to the half-life of peptides in the body. Therefore, the study of peptide stability in the gastrointestinal tract is also crucial for the development and utilization of its oral formulations. In the follow-up study, human intestinal biopsies, obtained from healthy controls, could be used to investigate the stability and biological activity of peptides [65]. Meanwhile, the peptides can be combined with drug delivery systems such as liposomes and nanoparticles, thus effectively addressing the issue of their metabolic stability [66].

Sika deer is a precious medicinal animal in China, but at present, the research on deer-derived products mainly focuses on velvet antler, while there are few related literature reports on deer bone, and the research on its biological activity is scarce and not in-depth, which limits the development and utilization of deer bone resources. Therefore, we extracted collagen peptide from deer bone to carry out experimental research. It was found that the extraction rate of collagen peptide from deer bone could reach 26.11% through preliminary research in a laboratory.

With the progress of science and technology and the continuous improvement of life quality, the demand for delaying physiological aging is increasing, and the consumption tendency and market expectation of anti-aging products have reached unprecedented heights, which has given birth to the urgent demand for diversified anti-aging products and services. Our results show that CP can improve the skin changes of aging mice induced with D-gal, enhance the hydration and antioxidant capacity of skin, and regulate the synthesis and degradation of collagen. It shows that deer bone collagen peptide may be used as the raw material of oral preparation to make functional food and play a role in delaying skin aging. The establishment of a single model in this paper can not fully simulate the complex environmental factors in real life. Nevertheless, our results can solve the urgent demand of anti-aging products to some extent, and provide a theoretical basis for the development and utilization of deer bone resources.

## 5. Conclusions

The experimental results show that the two kinds of deer bone collagen peptides are mainly peptides that constitute type Ⅰ collagen, and their peptide segments are different. Deer bone collagen peptide can promote skin hydration of aging mice by regulating moisturizing related factors, improve oxidative stress of aging mice skin by promoting the activity of antioxidant-related enzymes, and promote collagen synthesis and inhibit its degradation by regulating the TGF-β and MAPK pathways. And the effect of fermented deer bone collagen peptide is better than that of non-fermented deer bone collagen peptide.

## Figures and Tables

**Figure 1 nutrients-16-01548-f001:**
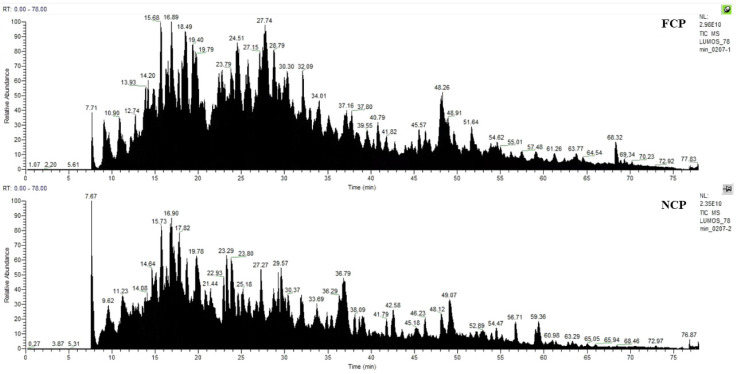
Mass spectrum Basepeak of FCP and NCP.

**Figure 2 nutrients-16-01548-f002:**
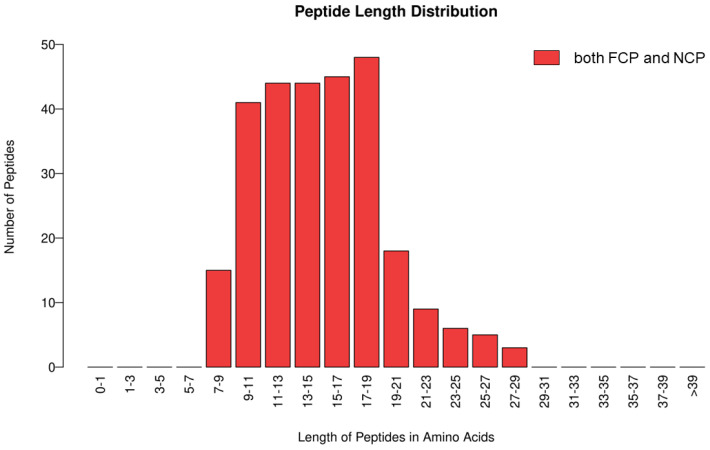
Distribution of peptide lengths contained in both FCP and NCP.

**Figure 3 nutrients-16-01548-f003:**
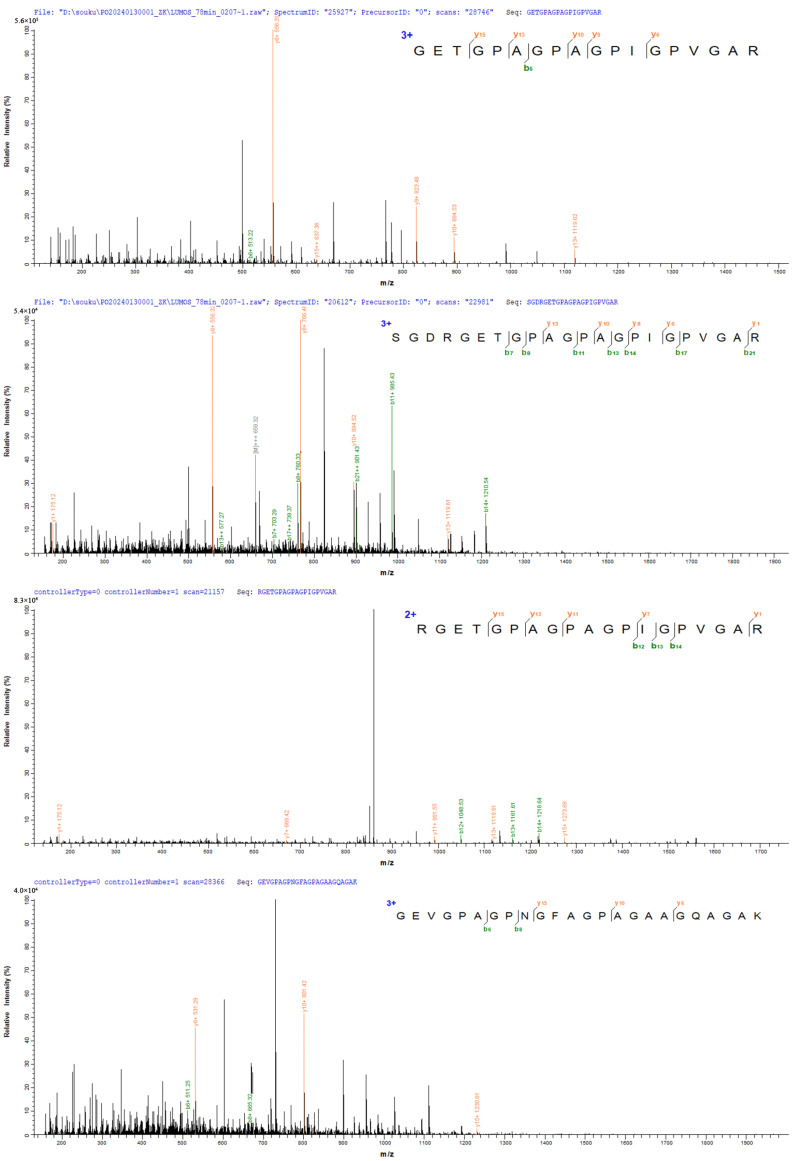
Peptides identified in FCP and NCP.

**Figure 4 nutrients-16-01548-f004:**
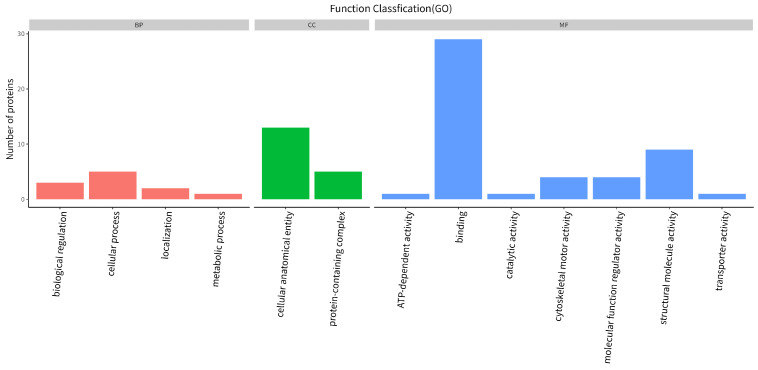
Functional annotation of differential protein GO.

**Figure 5 nutrients-16-01548-f005:**
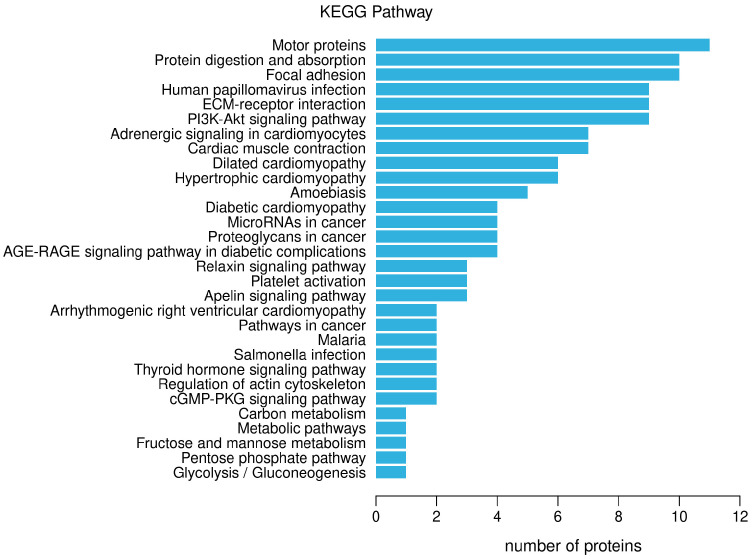
Notes on the KEGG pathway of differential proteins.

**Figure 6 nutrients-16-01548-f006:**
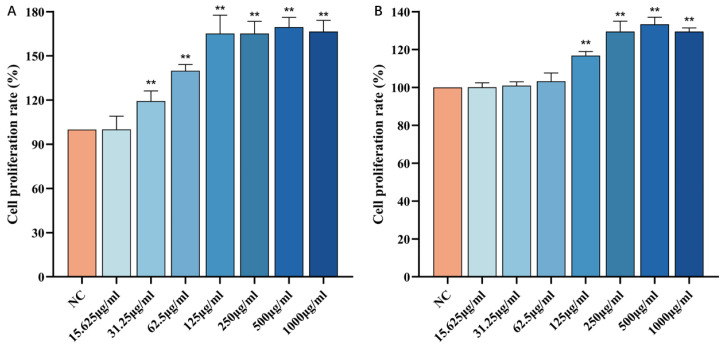
Effect of CP on the viability of HSF cells: (**A**)—Effect of FCP on the viability of HSF cells. (**B**)—Effect of NCP on the viability of HSF cells (*n* = 3). Note: Compared with the normal control group, ** *p* < 0 01.

**Figure 7 nutrients-16-01548-f007:**
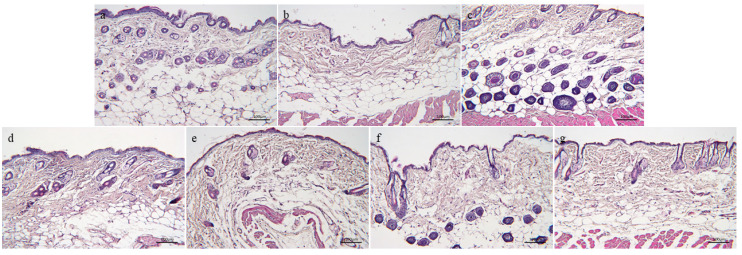
H&E staining of skin tissue of aging mice: (**a**)—Normal control group. (**b**)—Model group. (**c**)—Positive drug group. (**d**)—FH group. (**e**)—FL group. (**f**)—NH group, and (**g**)—NL group (*n* = 3).

**Figure 8 nutrients-16-01548-f008:**
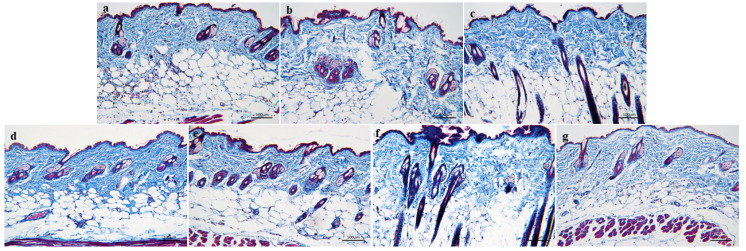
MASSON staining of the skin tissue of aging mice: (**a**)—Normal control group. (**b**)—Model group. (**c**)—Positive drug group. (**d**)—FH group. (**e**)—FL group. (**f**)—NH group, and (**g**)—NL group (*n* = 3).

**Figure 9 nutrients-16-01548-f009:**
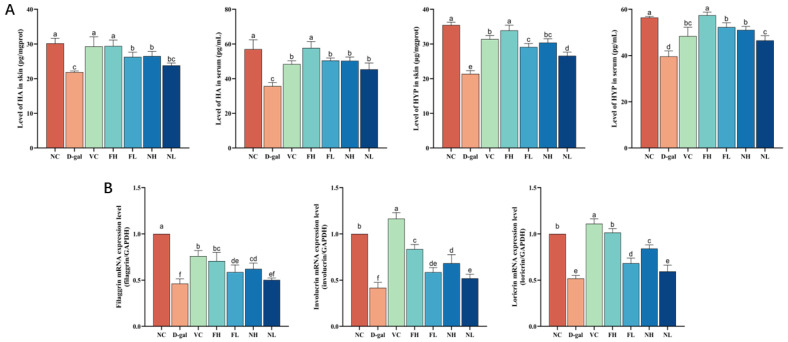
Effect of CP on skin hydration in aging mice: (**A**)—Effect of CP on HYP and HA content in skin and serum of aging mice. (**B**)—Effect of CP on the expression of filaggrin, involucrin, and loricrin genes in the skin of aging mice ($\bar{x}\pm s,\;n=3$). Note: Different letters indicate statistical difference between groups (*p* < 0.05).

**Figure 10 nutrients-16-01548-f010:**
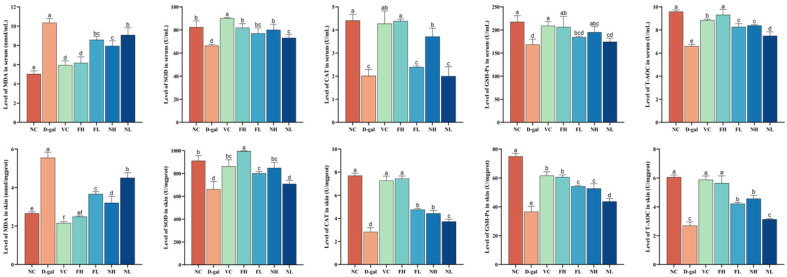
Effect of CP on the content of MDA, SOD, CAT, GSH-Px, and T-AOC in the skin and serum of aging mice ($\bar{x}\pm s,\;n=3$). Note: Different letters indicate statistical difference between groups (*p* < 0.05).

**Figure 11 nutrients-16-01548-f011:**
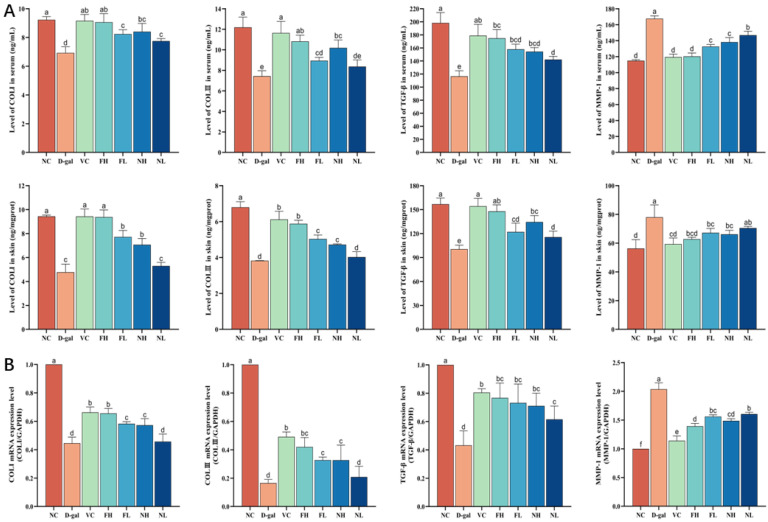
Effect of CP on collagen production ability of aging mice skin: (**A**)—Effects of CP on the content of COLⅠ, COLⅢ, MMP-1, and TGF-β in the skin and serum of aging mice. (**B**)—Effects of CP on expression of COLⅠ, COLⅢ, MMP-1, and TGF-β in the skin of aging mice ($\bar{x}\pm s,\;n=3$). Note: Different letters indicate statistical difference between groups (*p* < 0.05).

**Figure 12 nutrients-16-01548-f012:**
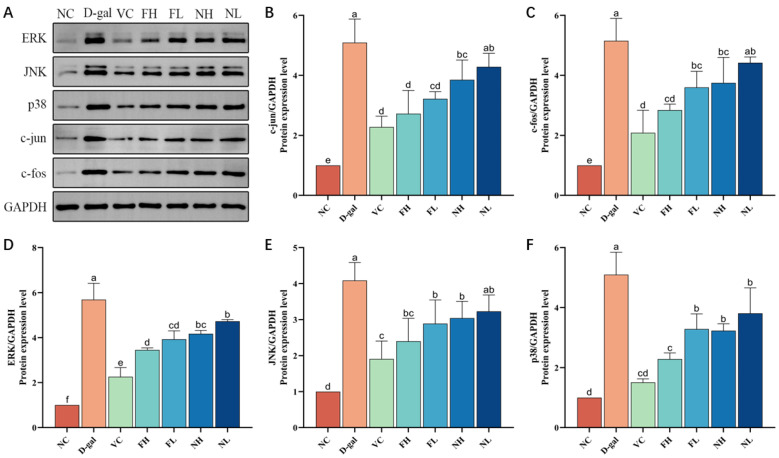
Effect of deer bone collagen peptide on the expression level of MAPK pathway-related proteins in skin of aging mice: (**A**)—Western blot test results of protein. (**B**–**F**)—Immunoblotting grayscale analysis ($\bar{x}\pm s,\;n=3$). Note: Different letters indicate statistical difference between groups (*p* < 0.05).

## Data Availability

The original contributions presented in the study are included in the article, further inquiries can be directed to the corresponding author.

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
