# Peer review of "Oral Administration of Deer Bone Collagen Peptide Can Enhance the Skin Hydration Ability and Antioxidant Ability of Aging Mice Induced by D-Gal, and Regulate the Synthesis and Degradation of Collagen"

_nutrients, 2024, doi:10.3390/nu16111548_

Round 1
Reviewer 1 Report
Comments and Suggestions for Authors
Understanding the mechanisms of aging and aging-related disorders necessitates the use of animal models for relevant studies. The authors wrote that their findings suggest deer bone collagen peptide could serve as a valuable raw material for functional health care products aimed at delaying skin aging. In this study, the main question is addressed to provide a more detailed clarification of the deer bone collagen peptides. The manuscript builds upon authors previous research on deer bone collagen, employing both fermented and non-fermented collagen preparations for further investigation in mice in vivo. This study adds data on the chemical composition of collagen fractions and identifies shorter peptides, as well as adds data to the search of skin anti-aging agents. This is main message which the paper addresses.
Permission to use experimental animals has been obtained from the national ethics commission. While the results section is extensive, the methods and materials lack sufficient details, leading to concerns and uncertainties regarding the kits and procedures employed. I recommend supplementing the text with a thorough description of the methods in the Methods section, as well evaluating the pros and cons of the accelerated aging method and need of bone fermentation in the Discussion.
The conclusions are partly consistent with the evidence presented (see Comments below).
Forty-three references are cited, and they are appropriate.
Many abbreviations are used, which can make it difficult for the reader. Therefore, it would be preferable to decipher abbreviations in the titles or in the section Methods.
I would like ask authors to revise the manuscript.
Comments:
Introduction: in the Introduction aging model must be explained at least in a few words. Clarification is needed regarding the accelerated aging model. Was the D-galactose (D-gal) model (dosage, timeline etc.) drawn from existing literature, or was it developed specifically for this study? Is it the most appropriate or commonly used model for studying skin aging? Subsequently, the Discussion should elaborate on the success of this aging model.
1. Line 95-96: It's unclear which precolumn and column of mass spectrometry were utilized.
2. Section 2.5: The rationale behind considering collagen peptide 400 mg/kg as a high dose and 200 mg/kg as a low dose in animal experiments is not explained. Where does this assumption originate from? What is optimal dose?
3. Line 133: Please provide a description of the mice (male, female, weight) and anesthesia used for mice at end-point.
4. Section 2.6: Details on how skin samples were prepared are needed, including whether feathers were shaved, and the size of the samples obtained.
5. Section 2.7; 2.8: Although analyzed biomarkers are well-known, their abbreviations should be clarified the first time they are mentioned in the text.
6. Section 2.8: Specify the producers and kit numbers for all ELISA kits used to study compounds. How absorbance or fluorescence was measured? Additionally, describe how serum (blood) and tissue samples were collected.
7. Section 2.10: The sentence "All groups were treated with lysate, and the protein was quantified with a BCA kit" lacks clarity regarding which groups are referred to. Furthermore, details about the primary antibodies used, including producer and dilution, are necessary.
8. Post hoc multiple comparison: Specify which post hoc multiple comparison method was used with one-way ANOVA.
9. Clarification on "skin": Define when "skin" refers to tissue, particularly in the context of skin homogenates.
10. Results: Provide clarity on what is meant by "collagen peptides" in the text. When FCP and when NCP?
11. Discussion. Add, what was found about the identified peptides in fermented collagen (FCP) and non-fermented collagen (NCP). Is there a plan to test shorter peptides in the future? Comparing the effects of deer bone collagen with the used skin anti-aging agents mentioned in the literature could add insight in perspectives utilize deer bone collagen. Does deer bone FCP or NCP collagen have any advantages? Why deer bones, wouldn't beef bones be more available? How much deer bones are needed to obtain enough collagen peptides? What was the extraction yield?
12. Conclusion. The conclusion does not provide a comparison of the difference or similarity of anti-aging effects between FCP and NCP and does not conclude which test shows the main mechanism of the anti-aging effect of CP. In Abstract there is a sentence "The results show that FCP and NCP are mainly peptides that constitute type Ⅰ collagen, and there are significant differences between them"'. What kind of differences?
Author Response
请参阅附件。

Reviewer 2 Report
Comments and Suggestions for Authors
The authors have undertaken an interesting and necessary topic, the interest of readers and scientists may be high.The biological requirements for newly discovered peptides to be used in anti-aging therapy in humans are usually very stringent.The work presents valuable effectiveness data confirming the amino acid composition, functional annotation, anti-free radical and moisturizing activity.
A).
The results are interesting and presented clearly.But they don't live up to the title. The paper describes the results of anti-aging effectiveness. The title suggests that these peptides can already be used in human anti-aging therapy, but only part of the research has been carried out.
There is a lack of tests and results proving the safe, stable, beneficial and key characteristics of newly discovered peptides:
1.Safety towards host cells: Peptides must be safe for humans and not cause harmful side effects or toxicity. Before being introduced into anti-aging therapy, peptides must undergo appropriate toxicological and safety tests, e.g. for the human skin fibroblast line; you can also calculate the safety index.
2. Targeted action: Peptides should have the ability to target specific molecules or receptors in the body that are associated with aging processes. This enables them to exert their intended anti-aging effect. Maybe the authors have data from Doking in silico?
3. Stability: Peptides must be stable enough to survive in the human body for an appropriate period of time. They must be resistant to digestive enzymes and not decompose quickly.
4. Tissue penetration: Peptides must be able to penetrate tissue barriers and reach target tissues and cells in the body. Special peptide delivery strategies, such as nanoparticles or carriers, may be required, as the authors suggest lyophilized administration.
5. Efficacy Peptides must be effective in combating aging processes and improving the health and function of the aging body. They must have a confirmed anti-aging effect at the cellular and molecular level - these tests and results are included in the work and beautifully presented by the authors.
The current title suggests that all key tests for new peptides have been performed, and only anti-aging effectiveness has been performed.
Therefore, either please do appropriate security and stability tests, or you will need to change the title, abstract and conclusions.
B)
Fig. 10 statistics - wouldn't it be better to compare each sample with each sample and with the control?
Author Response
请参阅附件。

Round 2
Reviewer 1 Report
Comments and Suggestions for Authors
The authors have considered all reviewer comments and made changes to the manuscript. I think the idea, the description of the experiments and the conclusions are now clearly understood. I have no critical comments and support the manuscript's acceptance for publication.
Author Response
Thank you again for your valuable time and effort in providing us with extremely valuable guidance to help us improve the quality of our articles.
Reviewer 2 Report
Comments and Suggestions for Authors
thank you for the authors' efforts in additional work, now the manuscript is complete.
Author Response

(The authors gave the same response as above.)
